# Nitric Oxide Trickle Drives Heme into Hemoglobin and Muscle Myoglobin

**DOI:** 10.3390/cells11182838

**Published:** 2022-09-12

**Authors:** Mamta P. Sumi, Blair Tupta, Arnab Ghosh

**Affiliations:** Department of Inflammation and Immunity, Lerner Research Institute, The Cleveland Clinic, Cleveland, OH 44195, USA

**Keywords:** apo-globins, heme, heme insertion, myoblasts, nitric oxide

## Abstract

Ever since the days of NO being proclaimed as the “molecule of the year”, the molecular effects of this miracle gas on the globins have remained elusive. While its vasodilatory role in the cardiopulmonary system and the vasculature is well recognized, the molecular underpinnings of the NO–globin axis are incompletely understood. We show, by transwell co-culture of nitric oxide (NO) generating, HEK eNOS/nNOS cells, and K562 erythroid or C2C12 muscle myoblasts, that low doses of NO can effectively insert heme into hemoglobin (Hb) and myoglobin (Mb), making NO not only a vasodilator, but also a globin heme trigger. We found this process to be dependent on the NO flux, occurring at low NO doses and fading at higher doses. This NO-triggered heme insertion occurred into Hb in just 30 min in K562 cells and into muscle Mb in C2C12 myoblasts between 30 min and 1 h, suggesting that the classical effect of NO on upregulation of globin (Hb or Mb) is just not transcriptional, but may involve sufficient translational events where NO can cause heme-downloading into the apo-globins (Hb/Mb). This effect of NO is unexpected and highlights its significance in maintaining globins in its heme-containing holo-form, where such heme insertions might be required in the circulating blood or in the muscle cells to perform spontaneous functions.

## 1. Introduction

Nitric oxide (NO) is a signal molecule and, amidst other functions, it plays a critical role in the regulation of vascular tone [1,2,3,4]. NO, generated by NOS enzymes [5,6,7,8] or produced by noncanonical means [9], is a central component of the NO-sGC-cGMP signal pathway and works by activating the soluble guanylate cyclase (sGC), an obligate heterodimer of α, β-subunits (sGC-αβ) [10,11,12]. cGMP produced by NO activation of sGC is known to transcriptionally activate fetal Hb (γ) gene expression and may also occur in primary human erythroblasts [13,14]. Likewise, NO is also known to transcriptionally stimulate myoglobin (Mb) gene expression [15]. While the heme prosthetic group is an essential component of these globins and is needed for globin functionality, our studies with globin heme maturations [16,17] have revealed that, in cells, there is a substantial amount of the heme-free globin or apo-globin (apo-Hb/apo-Mb), and this was previously unappreciated. Moreover, NO can directly cause heme insertion into the sGCβ subunit to cause its heterodimerization and subsequent activation [18,19].

In light of these finding, we speculated that the NO-driven globin synthesis and subsequent heme insertion/maturation can also have a translational component, wherein heme insertion into a pre-existing globin maybe possible. We thus studied the NO-driven globin heme insertion/maturation using transwell co-cultures of apical NO-producing cells and basal Hb/Mb-expressing cells. We document here that NO generated from both eNOS or nNOS can drive heme insertion into pre-existing fetal (K562) hemoglobin (Hbγ). A similar effect occurs in C2C12 muscle myoblasts, where the NO generated from nNOS triggers heme into muscle cells, and this may corroborate with physiological events ocurring at sites of spontaneous activity.

## 2. Material and Methods

### 2.1. Reagents

All chemicals were purchased from Sigma (St. Louis, MO, USA) and Fischer chemicals (Fair Lawn, NJ, USA). NOS inhibitors 1400 W, L-NMMA and NO scavenger, and Carboxy-PTIO were obtained from Sigma. Transwell inserts were purchased from Cell Treat Scientific Products (Pepperell, MA, USA). Stable cell lines of HEK293 expressing eNOS or nNOS were obtained from Dr. Dennis Stuehr’s lab, and these were originally gifted by Prof. Solomon Synder (Johns Hopkins, Baltimore, MD, USA). C2C12 and K562 cells were purchased from American Type Culture Collection (ATCC; Manassas, VA, USA). Antibodies specific to Mb, Hbα, or Hbγ were obtained from Santa Cruz BioTech (Dallas, TX, USA), while eNOS and nNOS antibodies were purchased from Cell Signaling Tech (Danvers, MA, USA).

### 2.2. Transwell Co-Culture and Growth/Induction of Cells

All cell lines were grown and harvested as previously described [17,20]. HEK cells expressing stable lines of eNOS or nNOS were first grown to confluency (50–60%); treated for 6 h with −/+ NOS inhibitors (20 µM of 1400 W, 10 µM of L-NMMA) −/+ NO scavenger, or Carboxy-PTIO (100 µΜ for a total time of 1.5 to 2 h, including 30 min incubation in co-culture with K562 cells); and then activated with Ca ionophore (10 µM) for an additional 30 min to induce nitric oxide (NO) generation. These cells were then co-cultured with basal K562/C2C12 cells for various lengths of time between 0 and 6 h (Figure 1). The cells were then harvested at various time points and the generated supernatants were assayed by Western blots for protein expression, globin heme insertion was analyzed by heme-stains, and NO generation (as nitrite) from NOSs was estimated by an ozone-based chemiluminescent assay [21]. Band intensities on Western blots or heme-stains were quantified using Image J quantification software (NIH).

### 2.3. Statistics

Statistics was performed on experiments that were repeated three times. Values were depicted as mean of *n* = 3 measures, ±SD. * *p* < 0.05 was considered significant, by one-way ANOVA.

## 3. Results

In order to determine the effect of NOS-generated NO on globin heme insertion, we first cultured HEK cells stably expressing eNOS or nNOS as apical cells in transwell inserts, while cultures of K562 were maintained in parallel. The corresponding NOS enzymes in the two stable lines were activated for NO generation by Ca ionophore for 30 min before these inserts were incubated by placing on top of K562 cells as transwell co-cultures (Figure 1 and Figure 2). Control experiments where the NOS lines were treated with NOS inhibitors 1400W or L-NMMA or with NO scavenger and Carboxy-PTIO (Appendix A) were performed in parallel [22]. These co-cultures were maintained for various lengths of time between 0 and 6 h, with both the apical and basal cells being harvested at specific time points and cell supernatants generated. As depicted in Figure 2, NO generated from both eNOS or nNOS caused a distinct heme insertion into Hbαγ in K562 cells (Figure 2A,B). There was an increase in the protein levels of Hbα/γ from 0 to 0.5 h of incubation with the activated apical NOS cells, but a still higher increase in the heme-levels of Hbαγ in the same window (a 1.1-fold Hbαγ protein increase compared with a 5.5-fold increase in Hbαγ heme levels caused by 30 min incubation of activated eNOS, and a 1.1-fold increase in Hbαγ protein compared with a nearly 7-fold increase in Hbαγ heme levels under a similar window of nNOS incubation, Figure 2B, lower panel), which suggests that heme-insertion occurred into the pre-existing apo-Hb. Control cultures of NOS lines that received NOS inhibitors (1400 W or L-NMMA) or those treated with Carboxy-PTIO (Appendix A) did not show such heme insertion into the Hbαγ, thereby proving that NO generated from the NOSs was the causative factor (Figure 2C,D and Appendix A). This NO-promoted heme insertion gradually decreased as NO levels accumulated from 0 to 6 h (Figure 2A,B), suggesting that such heme insertion was most effective at low NO doses (Figure 2C,D). We, however, evidenced some lowering in NOS protein levels at the 6 h period, which may have been because stable lines in G418 free media, when co-cultured for a long time, may cause lowering of NOS protein expression, due to deactivation of the plasmid DNA (Figure 2A). These data suggest that NO generated from NOS enzymes can cause a quick heme insertion into the pre-existing apo-Hb present in K562 cells [17]. We performed similar experiments with a combination of HEK nNOS and mouse C2C12 myoblasts, where Ca ionophore activated nNOS stables, meaning NO was incubated with C2C12 myoblasts, and heme insertion into the Mb was studied within a 0–6 h window. As depicted in Figure 3, we witnessed a similar NO-promoted heme insertion into muscle Mb, which was at a maximum at 1 h, before fading at higher NO doses. Again, in the C2C12 cells, a great majority of Mb is in the heme-free or apo state [16] and heme insertion into apo-Mb readily occurs at low NO doses. Together, our results confirm that low NO doses cause a direct heme insertion into the apo-globins, and this insertion primarily seems to occur post-translationally as there is little increase in the globin protein levels.

## 4. Discussion

While NO has long been postulated to bring about better oxygenation, the one explicit way by which this can happen is by increasing the heme-containing Hb levels in the blood, such that it can bind and deliver more oxygen [23]. The implications of these findings suggest that, in the vasculature, the vasodilatory effect of NO [23,24] can also be coupled with an increase in blood Hb levels. The latter effect is a novel outcome of our current findings and may happen simultaneously with vasodilation. Our data also suggest that NO can increase globin gene synthesis at the transcriptional level (Figure 2A,B), and this has been implicitly depicted in earlier studies [14], while post-translationally, it can cause heme insertion into the apo-globins, and thus can be implicated to be working at both levels. While the observed effects of NO are caused by low NO levels, which cause heme insertion into the globins, high NO levels actually inhibit heme insertion into Hb, as we determined earlier. Here, the optimum NO donor concentrations (from NOC-18) were close to 125 µM, which caused inhibition of Hb heme [25]. Previous studies also found that NO generated from iNOS inhibits its own heme insertion and subsequent dimerization, when NO accumulates post 12 h of iNOS induction in mouse macrophages (RAW 264.7), but not in the presence of NOS inhibitor, L-NAME [26]. Moreover, in our current study, the heme insertion we observe into Hb or Mb typically occurs in the low NO range (0–4 µM, as depicted from nitrite values, Figure 2C and Figure 3C), while higher nitrite values of >8 µM result in lowered heme insertion. Thus, it can be inferred that NO plays a dual role of promoting heme insertion at lower doses while impeding heme insertion at higher doses (Figure 2 and Figure 3).

Mechanistically, we do not understand how such NO-mediated globin heme insertion can occur, but may require the GAPDH-hsp90 nexus as these are proteins needed for the transport and delivery of heme into the globins [27]; at this stage, however, it would be rather speculative to suggest this, and further work is needed. There can be many scenarios where such NO-mediated globin heme insertions can occur. The most common of them can be in pathologies including tumors where there is inflammation and NOS enzymes are upregulated [28,29]. Here, NO-mediated Hb/Mb heme insertion would be a possible outcome to heme-mature these globins, such that the harmful oxidant effect can be countered with heme-containing globins [15,30]. Other scenarios may include the formation of a neuromuscular junction, where neuronal NO plays a critical role [31], and here, the resident Mb heme-maturation can be significant to improve the muscle tone. NO-mediated buildup of muscle tone during physical exercise may also involve the NO-mediated Mb heme insertion process. Moreover, during early developmental stages, the NO-mediated globin heme insertion can be a critical event [32,33]. Lastly, when NO is used as a inhalation therapy [34], such NO-mediated processes are likely to be in play, where it can increase insertions of globin heme. However, at this stage, it is difficult to corroborate these findings with actual physiological events, but there can be many such instances where these are likely to happen and our current findings, which are thought provoking, can allow further investigations into these critical processes.

## Figures and Tables

**Figure 1 cells-11-02838-f001:**
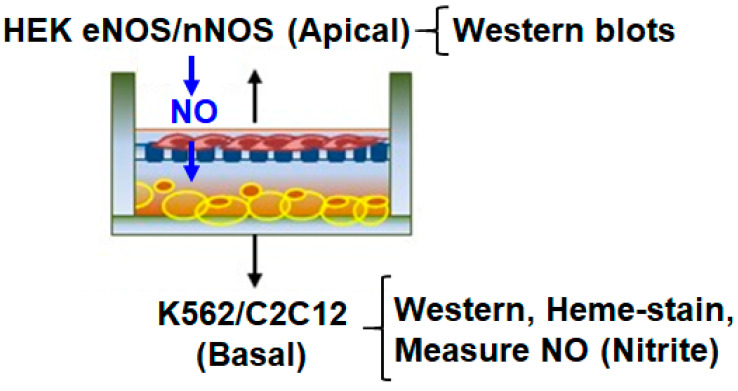
Representation of a transwell co-culture. HEK cells expressing stable lines of eNOS or nNOS were activated with Ca ionophore to induce nitric oxide (NO) generation, −/+ NOS inhibitors (1400W/L-NMMA), and then co-cultured with basal K562/C2C12 cells for various lengths of time between 0 and 6 h. Expressions of the globins (Hb/Mb) were assayed by Western blots, globin heme insertion was analyzed by heme-stains, and NO generation (as nitrite) from NOSs was estimated by an ozone-based chemiluminescent assay.

**Figure 2 cells-11-02838-f002:**
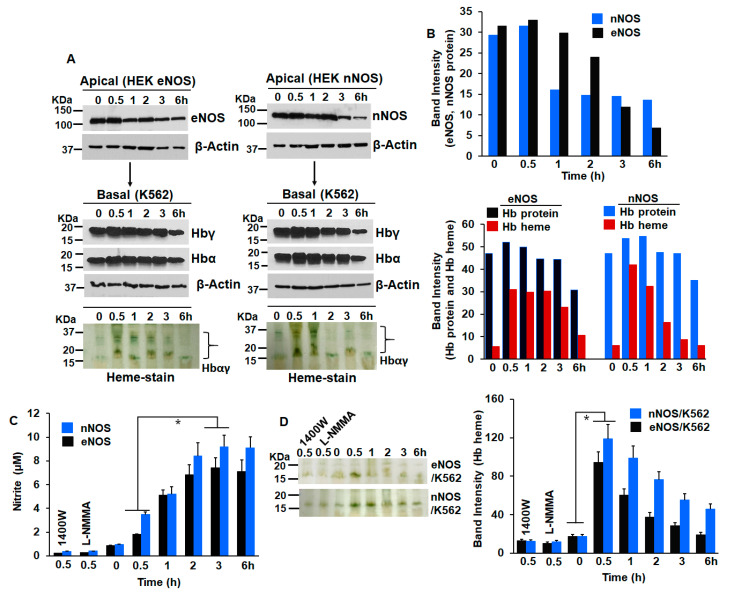
NO induces heme-insertion into fetal Hb. Stable lines of HEK cells expressing eNOS/nNOS were cultured and activated with Ca ionophore for 30 min, −/+ NOS inhibitors (1400 W/L-NMMA) before being co-cultured with K562 cells in a transwell for various lengths of time between 0 and 6 h. The cultures were then harvested and the generated supernatants were assayed for protein expression by Western blots, Hb heme by heme-stain, and eNOS/nNOS-generated NO (as nitrite) by an ozone-based chemiluminescent assay. Panel (**A**) Protein expression of eNOS/nNOS, Hbα/γ, and Hb heme-stains, as indicated. Panel (**B**) Corresponding densitometries of NOSs and Hb protein or heme levels, as depicted in panel A. Panel (**C**) NO estimation as nitrite by a chemiluminescent assay. Panel (**D**) Left, representative heme-stains of Hb, as indicated. Right, mean densitometries of heme-stains from three independent experiments. Values depicted are mean *n* = 3, ±SD. * *p* < 0.05, by one-way ANOVA. Molecular weight markers (KDa) are depicted at the left of gel bands throughout the figure legends.

**Figure 3 cells-11-02838-f003:**
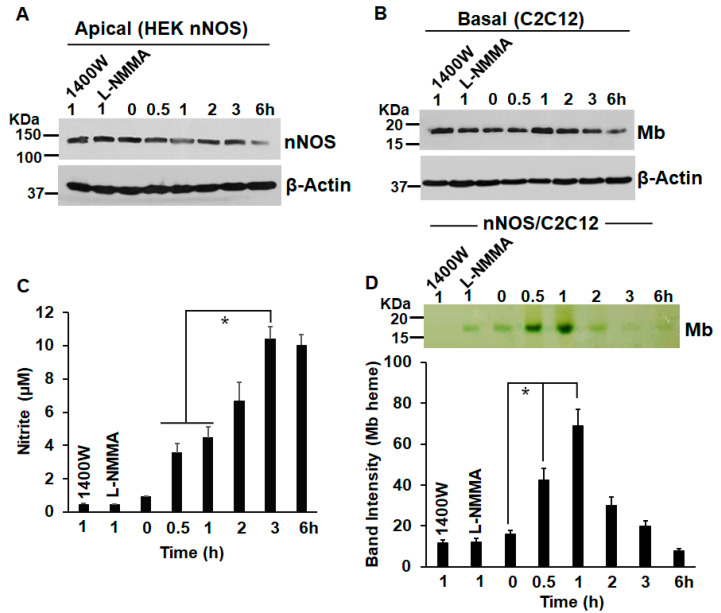
NO induces heme insertion into muscle Mb. Stable lines of HEK cells expressing nNOS were cultured and activated with Ca ionophore for 30 min, −/+ NOS inhibitors (1400 W/L-NMMA) before being co-cultured with C2C12 muscle myoblasts in a transwell for various lengths of time between 0 and 6 h. The harvested cultures were assayed similarly as described in Figure 2. Panel (**A**,**B**) Protein expression of nNOS and Mb, as indicated. Panel (**C**) Generated NO estimated as nitrite by a chemiluminescent assay. Panel (**D**) Upper, representative heme-stains of Mb, as indicated. Lower, mean densitometries of heme-stains from three independent experiments. Values depicted are mean *n* = 3, ±SD. * *p* < 0.05, by one-way ANOVA.

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
