# Peer review of "Nitric Oxide Trickle Drives Heme into Hemoglobin and Muscle Myoglobin"

_cells, 2022, doi:10.3390/cells11182838_

Round 1

Reviewer 1 Report (Previous Reviewer 1)

The authors responded to my concern satisfactory. No further comments.

Reviewer 2 Report (Previous Reviewer 2)

The manuscript has been significantly improved and now warrants publication in this journal.

This manuscript is a resubmission of an earlier submission. The following is a list of the peer review reports and author responses from that submission.

Round 1

Reviewer 1 Report

The authors showed, using the transwell co-culture system, that NOS inhibition suppresses NO production and heme insertion into heme proteins, including hemoglobin and myoglobin. This is a novel and interesting study but some issues must be addressed. My comments are as follows.

Major comments:

1.      Does pretreatment of cells with carboxy-PTIO, a NO scavenger, suppress the increase in heme levels of globins due to Ca ionophore addition? I think such data are very important in making a conclusion about the influence of NO itself. This is because products produced by NOS activation include not only NO but also citrulline, NADP and so on. In other words, there is no data that denies the influence of products other than NO.

2.      The authors’ group previously reported that NO inhibits heme insertion into hemoglobin (Free Radic Biol Med. 2010;48:1548-1558). This finding is the exact opposite of the current results. How do the authors explain this contradiction? Please discuss in the section Discussion.

3.      In the section Material and Methods, there is no description on how statistical analysis was performed. I do not understand what comparison results the symbols (asterisks) in Figure 2C and Figure 3C represent.

4.      Figure 2C and Figure 3C are very confusing. Is it correct to understand that the 6 columns on the right (0 to 6h) are in the absence of NOS inhibitors and only the 2 columns on the left (0.5 and 0.5) are in the presence of NOS inhibitors? If so, it should not be connected with a polygonal line. Rather, it would be better to make separately a 6 (for Figure 2: nNOS, eNOS, nNOS+1400W, eNOS+1400W, nNOS+L-NMMA, eNOS+L-NMMA)- or 3 (for Figure 3: nNOS, nNOS+1400W, nNOS+L-NMMA)-column bar graph.

5.      Please quantify heme-stains in the presence and absence of NOS inhibitors in Figure 2D and Figure 3D. Quantified graphs help readers understand.

6.      Page 5, lines 140-141: In this manuscript, there is no data showing that NO increases globin gene synthesis. What is the basis for “Our data also suggests that NO can increase globin gene synthesis at the transcriptional level”?

7.      Page 5, lines 148-149: The authors refer to iNOS (ref. 25) as a scenario in which NO-mediated heme insertion occurs. However, NO production by iNOS is much greater than that by eNOS and nNOS. Would a large amount of NO act as a heme insertion inducer? Of note, the authors state that NO-induced heme insertion is most effective in low doses (page 3, lines 99, 110-111).

Minor comments:

1.      Page 2, 2.1. Reagents: Where did the authors get K562 cells from?

2.      Page 2, lines 57-58: Please include city/state name for Santa Cruz and for Cell Signaling Tech.

Author Response

Hi,

Please see attached file for response to reviewer 1.

Arnab Ghosh

Reviewer 2 Report

Summary: In this manuscript, the authors tested their hypothesis that NO has a post-translational effect, specifically heme incorporation, on globins. Using transwell co-culture of NO-generating cells with either erythroid (for hemoglobin expression) or myoblasts (for myoglobin expression), the authors monitored heme incorporation into the globins as the cells are exposed to NO. Significantly, the authors noted that NO drove heme into the globins at low NO concentrations at early time points. This manuscript is relevant to and timely for the field, of great interest to those in the field, and uses a method (transwell co-culture) that is appropriate to test their hypothesis.

General comments:

1.     The authors note in the abstract, and emphasize throughout the manuscript, that heme insertion is “happening at low NO doses and fading at higher doses.” The most heme incorporation is seen at early time points when the cells have been exposed to less NO, so the reviewer could agree with the first part of their interpretation of the data if the authors, perhaps with some references, convince us that this is in fact a low dose of NO and can differentiate this from a burst, especially in contrast to a “trickle”. However, the “higher doses of NO” are cells being exposed to NO for longer time periods, so more NO in total. The experimental design does not control the initial doses of NO, so the reviewer feels like the authors need to be clearer in their explanations/definitions of doses in contrast to exposure time.

2.     The globin protein levels decrease over time at a very similar rate as the heme levels, so couldn’t the “fading at higher doses” be related to protein degradation rather than heme incorporation? If the authors disagree, perhaps a new way for presenting the data in figure 2B (that includes error bars) would be helpful to explain their interpretation?

3.     A previous report by the authors suggests that upwards of 60% of hemoglobin is in the apo form; here, the 5.5- to 7-fold increase in heme upon exposure to NO could suggest by my back of the envelope calculations that all the available apo-hemoglobin could be acquiring heme, so there would be little apo-protein available after the initial time point. Thus, any further heme incorporation would not be observed as the cells are being exposed to NO over time. Could the authors address this either with additional experiments (as they have used previously) or discussion within the manuscript? 

4.     The manuscript has many grammatical errors.

5.     The manuscript uses inconsistent nomenclature that makes it difficult to interpret. Specifically, the apical cells could consistently be referred to as “HEK-eNOS” and “HEK-nNOS” (or something similar) throughout. This became particularly confusing in figure 2B (lower) when upon initial glance it appeared as though nNOS or eNOS protein levels were being divided by Hb protein levels to present a ratio of protein levels. Also, in figure 2D, yet another way of naming the cells was used. Lastly, referring to the cell lines as “NOS stables” is very colloquial; the language of the text would be enhanced by improving the consistency of the nomenclature as above.

Specific comments:

1.     Line 41, page 1. Do the authors mean to state a that there is a post-translational component?

2.     Figures 2 & 3. The heme stain images are difficult to see, especially in figure 3; can the image quality be improved?

3.     Figures 2A and 2D. The heme stains in 2A show the dimer, while 2D does not, so it is difficult to conclude that these are showing the same results, particularly because the quantification of the stain in 2D does not appear to presented anywhere?

4.     Figure 2C. The data in 2C might be better represented by a bar graph, box plot, or scatter plot; it is not correct to connect with a line the data with the inhibitors with the data without the inhibitors. 

5.     Figure 3. This figure would be dramatically improved by adding the quantification of the data as seen in figure 2B. 

Author Response

Hi,

Please see our responses to Reviewer 2 in the attached file.

Thanks

Arnab Ghosh

Round 2

Reviewer 1 Report

The issues raised in my previous review have been largely addressed, but there are still serious concerns.

Major comments:

1.      What can be claimed from the results of NOS inhibitors is not the same as what can be claimed from the results of NO scavengers. As I previously mentioned, NOS inhibition decreases citrulline and NADP production. If the authors claim that “NO can effectively insert heme into globins”, data must be shown that Ca ionophore-promoted heme insertion is inhibited by NO scavenger(s) or that heme insertion is stimulated by low-dose NO donor(s). The authors’ opinion “These do not promote heme-insertion into the globins as heme-insertion was still inhibited with longer incubation time points (3-6h) and with increasing NO accumulation” is just a hypothesis.

3.      I still do not understand the details of the statistical method. One-way ANOVA is an omnibus test statistic and cannot tell which specific groups are different from each other. Did the authors perform a post-hoc test?

7.      As this study focused on eNOS- or nNOS-derived NO, there is a sense of incongruity in citing a paper on iNOS as a reference. iNOS is not the only NOS isoform that is upregulated in cancer, and I think paper(s) that also mention eNOS and nNOS are more suitable; for example, Cancer Res. 1995;55:727-730.

Author Response

Hi,

Please see the attached response to the questions raised.

Thanks

Arnab Ghosh
